# Venetoclax with Hypomethylating Agents in Newly Diagnosed Acute Myeloid Leukemia: A Systematic Review and Meta-Analysis of Survival Data from Real-World Studies

**DOI:** 10.3390/cancers15184618

**Published:** 2023-09-18

**Authors:** Andrealuna Ucciero, Federico Pagnoni, Lorenza Scotti, Alessia Pisterna, Francesco Barone-Adesi, Gianluca Gaidano, Andrea Patriarca, Monia Lunghi

**Affiliations:** 1Hospital Pharmacy AOU Maggiore della Carità, 28100 Novara, Italy; 2Department of Translational Medicine, Università del Piemonte Orientale, 28100 Novara, Italy; 3Division of Hematology, Department of Translational Medicine, University of Eastern Piedmont and AOU Maggiore della Carità, 28100 Novara, Italy

**Keywords:** acute myeloid leukemia, hypomethylating agents, real world studies, survival, venetoclax

## Abstract

**Simple Summary:**

The association of venetoclax (VEN) with hypomethylating agents (HMAs) such as azacitidine (AZA) and decitabine (DECI) significantly improved the outcome of patients with newly diagnosed acute myeloid leukemia (AML) who are not eligible for intensive chemotherapy. However, it is not clear how applicable the results of clinical trials are in a real-world setting. For this reason, we conducted a systematic review and meta-analysis of real-world studies on this type of AML patient. Overall, AML patients treated with VEN+HMAs had a lower survival rate than was reported in the pivotal VIALE-A trial. Results slightly increased when the analysis was restricted to the studies using VEN+AZA as treatment. Future studies are warranted to investigate whether this discrepancy is due to the different characteristics of enrolled patients or to a non-optimal adherence to therapy.

**Abstract:**

In recent years, the association of venetoclax (VEN) with hypomethylating agents (HMAs) significantly improved the outcome of patients with newly diagnosed acute myeloid leukemia (AML) who were unfit for intensive chemotherapy and became the standard of care after the publication of the pivotal RCT VIALE-A. However, it is still not clear to what extent the results observed in the VIALE-A apply to a real-world setting. For this reason, we carried out a systematic review and meta-analysis of real-world studies on newly diagnosed patients with AML, ineligible for intensive induction chemotherapy, receiving first-line VEN+HMA. We then compared their results in term of survival with those from the VIALE-A. Kaplan-Meier curves were extracted from all included studies and individual survival data was reconstructed. We then estimated a pooled survival curve and compared it with the results of the VIALE-A using the log-rank test. We also conducted a secondary analysis including only studies considering VEN plus azacytidine (AZA) as treatment, as this was the schedule originally used in the VIALE-A. Nineteen real-world studies met the inclusion criteria and were included in the systematic review. Most of them reported a worse survival than the VIALE-A. The pooled survival curve was similar to that reported in the VIALE-A during the first three months of treatment but diverged thereafter (*p*-value = 0.0001). The pooled median survival among the real-world studies was 9.37 months (95%CI 8.81–10.5), substantially lower than that reported in the VIALE-A (14.7 months; 95%CI 11.9–18.7). Results slightly increased when the analysis was restricted to the studies using VEN+AZA as treatment (median survival: 11.5 months; 95%CI 10.2–14.8). Survival of newly diagnosed AML patients treated with VEN+HMAs in a real-world setting seems to be lower than previously reported in the VIALE-A, while the effect of VEN+AZA is more in line with expected results. Future studies are needed to evaluate whether this apparent discrepancy is due to the different characteristics of enrolled patients or to a non-optimal adherence to therapy, and whether alternative regimens can provide better results in terms of safety and effectiveness.

## 1. Introduction

Acute myeloid leukemia (AML) is a rapidly progressing neoplasm characterized by infiltration of the blood, bone marrow, and other tissues by clonal myeloid blast cells. AML can occur at any age. While it represents an important share of childhood cancers (it includes about 25% of childhood leukemias), its incidence increases with age, with a median age at onset of 68 years. Indeed, more than 60% of AML occurs in patients over the age of 55 and it represents the most common form of acute leukemia in adults [1]. In western countries, age-adjusted incidence rates of AML vary between 3 and 6 cases per 100,000 person-years, with males having a 1.5-fold increased risk of disease compared to females. Worldwide, this condition is responsible for more than 80,000 deaths every year, and the trend is expected to substantially increase in the coming years [2]. In the United States, AML is responsible for only 1% of new cancer diagnoses but causes about 60% of leukemic deaths, with 5-year survival being below 30% [3]. The etiology of AML is heterogeneous and only partially known. Genetic factors, including germinal predispositions (e.g., the CCAAT/enhancer binding protein α—CEBPA and DEAD-Box Helicase 41—DDX41), Li Fraumeni syndrome (TP53), and bone marrow failure syndromes, are associated with both the onset and response to therapy for this condition [3]. Moreover, about 30% of AML are secondary to a previous hematological disorder, including myelodysplasia, or related to past therapies (e.g., following radiotherapy or chemotherapy with alkylating agents or topoisomerase II inhibitors). These secondary and therapy-related AML are associated with a particularly poor prognosis [3]. Signs and symptoms of AML are mainly caused by the replacement of bone marrow with neoplastic cells and include fatigue, shortness of breath, tachycardia, and exertional dyspnea due to the anemia caused by the decrease in red blood cells. In addition, bruising and bleeding can often occur due to the reduction of platelets (spontaneous bleeding may occur in severe cases, including intra-abdominal or intracranial hematomas), together with an increased risk of both bacterial, viral, and fungal infections caused by the lowering of the number of white blood cells. Other nonspecific and general signs and symptoms include fever, weight loss, enlarged spleen and liver, and widespread bone, joint, and abdominal pain [1]. The prognosis of AML is strongly associated with age at onset of the disease. The 5-year overall survival drops from more than 60% for patients aged up to 50 years to less than 10% for those aged 65 years or more at the time of diagnosis. Other important prognostic factors include the general performance status of the patient (usually measured through the Eastern Cooperative Oncology Group—ECOG—performance scale) and their cytogenetic features. In particular, different studies suggest that patients carrying mutations of TP53 and fms-like tyrosine kinase 3 (FLT3) generally have a poor prognosis, while those with mutations of Nucleophosmin-1 (NPM1) and CEBPA have a more favorable outcome [1]. Standard treatment for AML includes intensive chemotherapy based on a combination of anthracycline and cytarabine, followed by allogenic stem cell transplant. However, AML patients are often not eligible for induction chemotherapy because of elderly age and the presence of comorbidities (e.g., congestive heart failure, chronic kidney impairment) [4]. These scenarios are typically associated with the worst prognoses. Furthermore, despite having had a reduction in mortality over the years, stem cell transplantation is still today a complex medical intervention that exposes the patient to multiple risks in the short to medium term. For example, it reduces the patient’s quality of life and autonomy, increases the risk of infectious complications, and can lead to fatal adverse events. Therefore, despite being the only curative solution to date, stem cell transplantation remains a possibility of treatment that is not viable in many cases due to the poor general health conditions of the patients [5]. In this set of patients, as an alternative to palliative treatment, hypomethylating drugs (HMAs) such as azacitidine (AZA) or decitabine (DEC) have long been considered the best therapeutic option, even if they are associated with a median overall survival of only 7–10 months (against a life expectancy of 2 to 8 months in the case of palliative treatment only) [6,7,8].

In recent years, novel treatments have become recently available for AML patients who are ineligible for induction chemotherapy and allogenic stem cell transplant. In particular, the introduction of venetoclax-based combination therapies has constituted a paradigm shift in the treatment of AML [9]. Venetoclax (VEN) is a target therapy drug belonging to the group of selective B-cell lymphoma-2 (BCL-2) inhibitors. The BCL-2 family includes multiple proteins (e.g., BCL2, BCL-XL, MCL1, BCL2A1) specifically involved in the intrinsic mitochondrial apoptotic pathway. As inappropriate cell survival and dysregulated apoptotic processes are one of the hallmarks of malignancies, this family of proteins has become a promising therapeutic target for cancer since its discovery more than 30 years ago [10]. The mechanism of action of venetoclax includes its binding to the BH3 domain of the BCL-2 protein, with subsequent release of proapoptotic proteins that cause death cell induction [11]. In AML, most of the leukemic cells display high levels of BLC2; thus, recent research has focused on the possibility of using VEN in this setting. While VEN, as a single agent, has showed quite low efficacy in the treatment of AML, different studies have found that its use in combination with other chemotherapy drugs is able to substantially reduce the number of leukemic cells and slow down the evolution of the disease [12]. In 2018, a phase IB study of 57 patients reported that treatment with VEN in combination with HMAs was associated with complete remission in about 60% of cases. Based on these results, the United States (US) Food and Drug Administration granted conditional approval for the combination of VEN and HMAs (VEN+HMA) for the treatment of patients 75 years and older, and for those who are otherwise ineligible for induction chemotherapy [13]. Thereafter, this combination was granted full approval in the US and Europe after the publication of the results of a randomized phase III clinical trial (VIALE-A). This study confirmed composite complete remission rates of more than 60% in patients receiving VEN+AZA and, more importantly, showed a substantial improvement of survival (14.7 vs. 9.6 months) in patients treated with VEN+AZA compared to placebo+AZA [14]. Recently, the results on the efficacy of VEN+HMAs have been confirmed by a meta-analysis of randomized controlled trials that reported a median survival of about 12 months in patients with new-onset AML who were ineligible for intensive chemotherapy [15]. Moreover, different studies showed the efficacy of venetoclax in patients with different types of mutations. In particular, the superiority of VEN+AZA vs AZA alone in terms of remission rates and overall survival has been reported in patients with mutations of NPM1, FLT3, isocitrate dehydrogenase (IDH) enzymes, and spliceosomes (SRSF2, SF3B1, U2AF1, ZRSR2) [16,17,18,19]. In patients with poor-risk cytogenetics and TP53 mutations, VEN+AZA was instead associated with better remission rates but not improved survival [3,19]. Recently, Pratz and colleagues have also shown that VEN+HMAs substantially slows time to deterioration of health-related quality of life (HRQoL), concerning, in particular, global health status, physical functioning, and cancer-related fatigue [20]. For all these reasons, therapy with VEN+HMAs is nowadays considered the primary therapeutic choice for AML patients who are unfit for induction chemotherapy.

Though the RCTs following the VIALE-A confirmed the results of this pivotal trial, a number of real-world studies that accumulated over the last few years reported a less clear benefit associated with the use of VEN+HMAs [14,21,22,23,24]. In particular, these studies reported a median survival ranging between 8.1 months (95% CI, 6.3–9.7) and 12.3 months (95% CI, 8.1–16.5), well below that of the VIALE-A trial. However, it should be considered that most of these studies had a limited sample size; thus, a thorough evaluation of effectiveness of venetoclax in a real-world setting is still lacking. For this reason, we carried out a systematic review and meta-analysis of real-world studies on newly diagnosed patients with AML, ineligible for intensive induction chemotherapy, receiving first-line VEN+HMA. We then compared their results in term of survival with those from the VIALE-A trial.

## 2. Materials and Methods

### 2.1. Search Strategy

We followed the Preferred Reporting Items for Systematic Reviews and Meta-Analyses (PRISMA) guidelines. Two search strategies were adopted. First, PubMed was searched from 1 January 2019 to 9 March 2023 using the following search string: Venetoclax [Title/Abstract] AND (acute myeloid leukemia [Title/Abstract] OR AML [Title/Abstract]). Although the VIALE-A trial was published in August 2020, we start the research literature in January 2019, as the first conditional approval of venetoclax from FDA was issued on 21 November 2018. Thus, some real-world studies (e.g., Winters 2019) had already been conducted in 2019, before the publication of VIALE-A trial. As a second search strategy, we used Scopus to find studies that had cited the VIALE-A trial [14].

### 2.2. Inclusion and Exclusion Criteria

The studies included in this systematic review met all the following inclusion criteria:Real-world studies (i.e., studies conducted out of an experimental setting);patients aged at least 18 years;previously untreated Acute Myeloid Leukemia (including also secondary AML and therapy-related AML); venetoclax used as first line of treatment in association with hypomethylating agents (azacitidine and/or decitabine);articles written in the English language.

Moreover, studies were excluded if they focused on relapsed/refractory acute myeloid leukemia, or if they did not provide results of a survival analysis. When estimates from different papers were based on data from the same source of patients, those based on the larger number of subjects (usually the most recent one) were chosen. 

### 2.3. Data Extraction and Synthesis

Two authors (FP and FBA) independently screened the articles to assess the studies’ eligibility for inclusion. Inconsistencies were resolved after a discussion. In case consensus could not be reached, a third author (AU) acted as an arbitrator. The following data were extracted from the included studies: first author name, year of publication, study design, number of centers involved, number of patients enrolled, age, sex, AML type, treatment administered, follow-up duration, and some clinical characteristics, such as European LeukemiaNet (ELN) classification, karyotype presentation, bone marrow blast count, hemoglobin, white blood cell count, platelet count, and reported adverse events (AEs).

### 2.4. Statistical Analysis

Kaplan-Meier (KM) curves reporting the overall survival among subjects treated with venetoclax plus hypomethylating agents were extracted from all included studies. From each curve, the individual survival data was reconstructed using the method described by Liu et al. [25] and previously used by Qin et al. [15] for pooling results from RCTs investigating the efficacy of VEN+HMAs. First, we extracted raw data coordinates from each survival curve using R software. Second, we reconstructed individual patient data (IPD) from the extracted data using the IPDformKM R package. For each curve, we evaluated the accuracy of the reconstruction using the tools included in that package. As suggested by Liu et al. [25], we considered a reconstruction to be sufficiently accurate when it had a small mean squared error value (<0.05), a mean absolute error (<0.02), and a maximum absolute error (<0.05), as well as a large *p*-value of the Kolmogorov-Smirnov test. Moreover, we graphically compared the reconstructed curves with the original ones. As a final stage, we estimated a pooled survival curve and compared it with the results of DiNardo et al. [14] using the log-rank test. Several secondary analyses were performed to assess the robustness of results of the main analysis. First, the pooled survival curve was estimated including only studies considering VEN+AZA as treatment, as this was the schedule originally used by DiNardo et al. [14]. Second, studies based on selected populations at higher risk of death (e.g., TP53-mutated AML, secondary AML, and therapy-related AML) were provisionally excluded from the analysis to assess their impact on the survival curve. Finally, the robustness of our results was evaluated through an influence analysis, where we iteratively estimated the median survival time after the exclusion of one study at a time (“leave one out” approach). The obtained estimates were then compared with the main results of our analysis and with those reported in VIALE-A to evaluate whether the results were driven by a particular study.

All tests were two-sided and performed at the 5% level of statistical significance. Statistical analyses were performed using R software (v 4.2.1 for IPDformKM package and v 4.3.2 for survival and survminer package R core team).

## 3. Results

The search identified 1032 unique papers. After title and abstract screening, 60 studies were retrieved and analyzed in full text. Among them, 41 were then excluded as they were not real-world studies (N = 10), did not schedule a venetoclax plus hypomethylating agent treatment (N = 12), did not provide a survival analysis result (N = 9), focused on relapsed/refractory acute myeloid leukemia (N = 5), did not use venetoclax as first-line treatment (N = 3), were not written in English (N = 1), or were based on the same source of patients of other included studies (N = 1). As a result, 19 studies met the inclusion criteria and were included in the analysis. Notably, even though the studies of Vachhani et al. [26] and Mathews et al. [27] were based on the same source of patients (the Flatiron Health database), the former included patients using VEN with any HMA, while the latter considered only patients using VEN+AZA. We then used the estimate of Vachhani et al. [26] for the main analysis and that of Mathews et al. [27] for the secondary analysis, restricted to VEN+AZA users. The process of article selection and the reasons for exclusion are shown in the PRISMA flowchart (Appendix A).

The characteristics of the included studies are displayed in Table 1 and compared with the VIALE-A trial. Most of the real-world studies considered in our systematic review were conducted in a single center, with the notable exception of Matthews et al. [27] and Vachhani et al. [26], which included a large number of centers as they have used the Flatiron database as a recruiting source. Overall, the real-world studies included a total of 1134 patients, with sample size for each study ranging between 13 and 169 patients, followed for a period ranging between 6 and 83 months. The percentage of males among patients ranged from 43% to 77%, and mean age at recruitment was between 60 and 79 years. These characteristics were not very different from those of patients recruited in the RCT of DiNardo (60% male, with a mean age of 76 years). 

In most of the studies, patients with normal karyotypes ranged between 32% to 42%, similar to DiNardo et al. [14] (44%). Notably, the study of Venugopal et al. [41], which was restricted to TP53-mutated patients, reported a much lower percentage of patients with normal karyotypes (3%) (Appendix A). In studies where this information was provided, patients with an intermediate ELN score ranged between 10% and 82%, and the average percentage of bone marrow blasts at the beginning of treatment ranged between 25% and 55%. Average levels of hemoglobin, white blood cells, and platelets instead ranged between 8 and 9 g/dL, 3 × 10^3^/µL and 15 × 10^3^/µL, and 34 × 10^3^/µL and 110 × 10^3^/µL, respectively. A direct comparison with the study of DiNardo regarding these characteristics was not possible, as this information was not reported in the publication, or was displayed in a different way (e.g., percentage of patients with anemia instead of average level of hemoglobin). Eleven studies also provided information on the frequency of adverse events during treatment with VEN+HMAs, namely grade 3–4 neutropenia (range among the different studies: 29–93%), thrombocytopenia (range: 14–90%), febrile neutropenia (range 13–80%), tumor lysis syndrome (range: 0–12%) [21,24,26,27,28,29,32,33,35,36,42]. The percentages of patients who discontinued venetoclax because of adverse events ranged in the different studies between 7% and 66% (Appendix A).

Figure 1 and Appendix A display KM curves of the different studies, showing that most of the real-world studies reported a worse survival than the RCT of DiNardo et al. [14]. Results of the log-rank test for the pairwise comparisons between single studies and DiNardo et al. [14] are reported in the supplementary material, with *p*-values ranging between <0.001 and 0.65 (Appendix A). The pooled survival curve reported in Figure 2 confirms this result. Its trend was similar to the curve reported by DiNardo et al. [14] during the first three months of treatment but significantly diverged thereafter (*p* value = 0.0001). The pooled median survival among the real-world studies was 9.37 months (95% CI 8.81–10.5), substantially lower than that reported by DiNardo et al. [14] (14.7 months; 95% CI 11.9–18.7). The median survival time slightly increased when the analysis was restricted to the only 6 studies using VEN+AZA as treatment (median survival: 11.5 months; 95% CI 10.2–14.8) (Figure 3) [27,28,31,34,38,42].

Other secondary analyses did not appreciably change the results. After exclusion of three studies based on subgroups of patients at higher risk of death (TP53-mutated AML, secondary AML, and therapy-related AML), median survival was 9.79 months (95% CI 8.87–10.8) (Appendix A). Finally, influence analysis shows that the provisional exclusion of each study did not substantially modify the estimate of the pooled median survival (results ranging between 9.34 and 10.23 in the different iterations), which remained much lower than that reported by DiNardo et al. [14] (Appendix A). 

## 4. Discussion

In recent years, the association of venetoclax with HMAs significantly improved the outcome of newly diagnosed AML unfit for intensive chemotherapy and became the standard of care in this subgroup of patients [13,14]. However, most of the real-world studies included in our review failed to reach the impressive results reported by the pivotal VIALE-A trial [14]. Indeed, although the trial of DiNardo et al. [14] reported a median overall survival of 14.7 months (95% CI 11.9–18.7), real-world studies showed substantially lower survival. With the only exception of Cherry et al. [31] and Mirgh et al. [36], which reported a median survival of more than 15 months, the other authors showed an overall survival ranging between 4 and 13 months [21,24,26,27,28,29,30,31,32,33,34,35,36,37,38,39,40,41,42]. As a consequence, the pooled median survival in our meta-analysis was 9.37 months (95% CI 8.81–10.5), significantly lower than that reported by DiNardo et al. [14]. This difference was also confirmed in the secondary analyses that we performed, in particular after we excluded studies enrolling subjects with worse prognoses (i.e., secondary, therapy-related, and TP53-mutated AML). While these results are quite clear, their interpretation is not straightforward. One possible explanation is that patients included in real-world studies were not directly comparable with those enrolled in the VIALE-A trial. While we observed that some characteristics such as gender, age, and karyotype were reasonably similar in the considered real-world studies and in the RCT of DiNardo et al. [14], we were not able to evaluate the prevalence of other important prognostic factors (e.g., ECOG performance status scale, baseline cytopenia grade, pre-treatment laboratory values, such as potassium, phosphate, uric acid, calcium, and creatinine) [43]. A possible heterogeneity of the clinical features among patients included in the different studies conducted in this setting was also suggested by Du and colleagues, who recently conducted a systematic review and meta-analysis of studies of AML patients treated with VEN+AZA [44]. Future research aimed at thoroughly evaluating the role of different factors in response to VEN, and allowing a better risk stratification of patients, is thus warranted. 

A second possible interpretation of the observed difference in survival between real-world studies and the VIALE-A trial could be due to the different duration and dosage of VEN treatment. Indeed, while the discontinuation rate of VEN due to the occurrence of adverse events was 24% in the VIALE-A trial, some real-world studies reported a frequency of more than 50% among their patients [21,36]. Interestingly, grade 3–4 neutropenia was, in general, reported more frequently among real-world studies than in the VIALE-A trial [29,32,33,36,42]. As survival was very similar among real-world studies and the VIALE-A trial during the first three months of therapy but diverged thereafter, suboptimal compliance to the original treatment schedule due to the occurrence of adverse events could be, in part, responsible for this phenomenon. The difficulty of maintaining the standard dose of VEN at 400 mg over time was underlined by different authors [21,29,33,34]. In particular, hematological toxicity and drug-drug interactions, due to the concomitant azole-based antifungal prophylaxis, were reported among the main causes of treatment reduction and interruption [21,29,33,34]. This aspect is especially relevant, as prolonged myelosuppression due to VEN+HMAs regimen requires antifungal prophylaxis with these drugs in the majority of patients [45]. Azole agents are CYP3A4 enzyme inhibitors and are thus able to affect VEN metabolism. A reduction in the metabolism of VEN can, in turn, lead to the hyper-dosage of this drug, with a subsequent higher risk of tumor lysis syndrome (TLS), a potential complication already reported in early chronic lymphocytic leukemia trials and in the VIALE-A trial [14,46]. To avoid this problem, some authors suggest reducing the dose of VEN by at least 75% before starting antifungal prophylaxis [45,47,48]. Others instead propose to stratify the dose of VEN based on the enzyme potency of the antifungal treatment (e.g., 70–100 mg/day with strong enzyme inhibitors and 200 mg/day with intermediate enzyme inhibitors) [45]. Future studies are thus needed to confirm the optimal schedule in maintaining VEN efficacy while reducing the occurrence of severe AEs. 

A third possible explanation of less favorable results reported in the real-world studies compared to the VIALE-A trial could be due to the type of HMA used in association with VEN. Indeed, the original schedule proposed by DiNardo et al. [14] was VEN+AZA, while most of the real-world studies also included patients using VEN+DEC. Interestingly, when we restricted our analysis to the studies using only VEN+AZA, results were closer to those of the VIALE-A trial, compared to the main analysis. However, it should be noted that this secondary analysis was based only on 6 studies out of 19 (365 patients out of 1196). Thus, the 95% CI of the estimated median survival was wide (survival: 11.5 months; 95% CI 10.2–14.8) and largely overlapping with the results of the VIALE-A trial (median survival 14.7 months; 95% CI 11.9–18.7). While it is possible that VEN+AZA is more effective than VEN+DEC, we think that our results are not strong enough to fully support such a statement and that further studies are needed to better investigate the association of VEN with AZA. Of note, different countries have been recently moving toward the approval of VEN+AZA rather than VEN+DEC. In Italy, after a period of two years under an early access program (provided by the National Law 648/1996), VEN has been fully approved in combination with AZA (but not with DEC), for the treatment of adult patients with acute myeloid leukemia who are ineligible for standard induction chemotherapy [49]. A similar decision has recently been taken in France and the UK as well [50,51]. On the other hand, in the US, venetoclax was granted accelerated approval, not only in combination with AZA, but also with decitabine or low-dose cytarabine [52]. 

This systematic review has some strengths. It is the first attempt to evaluate the survival of newly diagnosed AML patients treated with VEN in a real-world setting. As most of the included studies were of limited sample size, the meta-analytic estimates provided in the present study are particularly valuable to thoroughly evaluate the effect of VEN. Notably, differently from other authors, we decided to focus our analysis only to patients with a new diagnosis of AML to obtain a more homogeneous set of patients and, thus, more interpretable results. Moreover, we used an established statistical method to extract and pool the survival curves of the single studies [25]. This rather new approach complements more traditional ones based on summary estimates (e.g., meta-analysis of relative risks); for example, allowing the evaluation of how the treatment’s effect changes over time. Our study also has different limitations, though. As we displayed in Appendix A, the original studies reported limited information on molecular profile of patients and their risk classification. For this reason, it was not possible to stratify our results for these important characteristics, something that would have provided more insights from our analysis. Another important limitation is due to the fact that real-world studies, such as those that we included in our analysis, typically do not have a control group to contrast with patients treated with VEN. For this reason, it was not possible to estimate a direct measure of effect (e.g., hazard ratio) of VEN+HMAs vs HMAs alone. 

## 5. Conclusions

In conclusion, survival of newly diagnosed AML patients treated with VEN+HMAs in a real-world setting seems to be lower than previously reported in the VIALE-A trial, while the effect of VEN+AZA is more in line with expected results. Future studies are needed to evaluate how much of this apparent discrepancy is due to the different characteristics of enrolled patients or to a non-optimal adherence to therapy, and whether alternative regimen schedules can provide better results in terms of safety and effectiveness. 

## Figures and Tables

**Figure 1 cancers-15-04618-f001:**
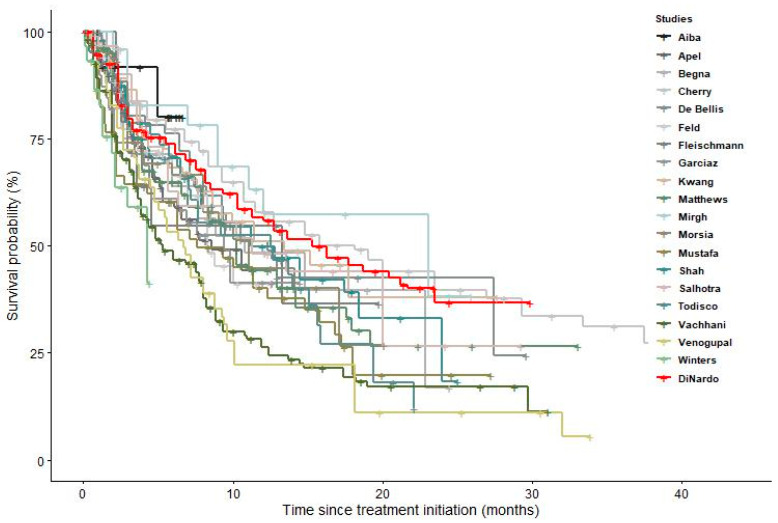
Survival curves of the included studies, compared with that of DiNardo et al. [14].

**Figure 2 cancers-15-04618-f002:**
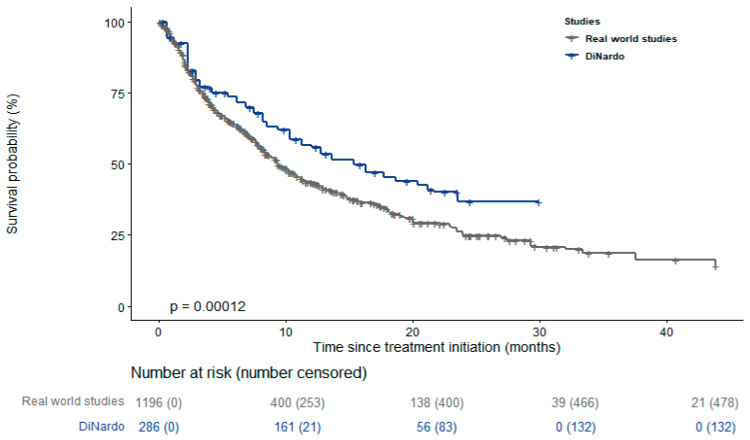
Pooled survival curve compared with that of DiNardo et al. [14].

**Figure 3 cancers-15-04618-f003:**
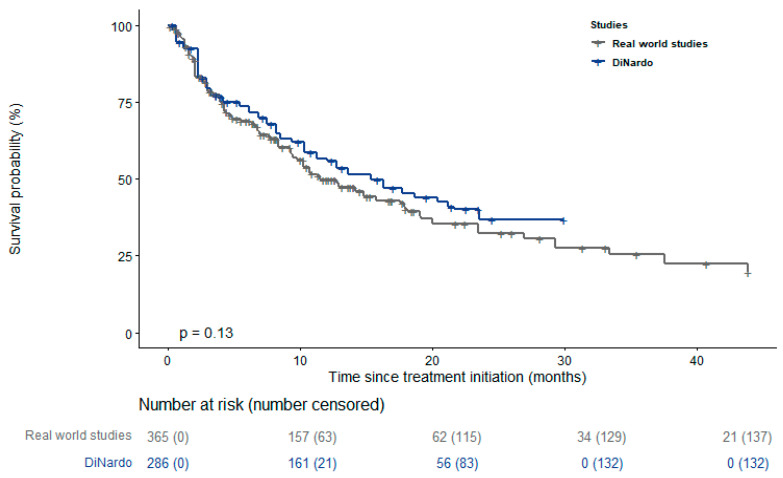
Pooled survival curve of the studies that consider VEN+AZA as treatment compared with that of DiNardo et al. [14] Only the studies of Aiba et al. [28], Cherry et al. [31], Garciaz et al. [34], Matthews et al. [27], Mustafa et al. [38], and Winters et al. [42] were considered in this analysis.

**Table 1 cancers-15-04618-t001:** Characteristics of the included studies, compared with those of DiNardo et al. [14].

Study	Number of Centers	N. Patients	Age— Mean (Range)	Males %	AML Type	Treatment	Follow Up—Months
Aiba, 2023 [28]	1	13	79 (72–86)	77	Any AML	VEN+AZA	6
Apel, 2021 [29]	11	133	77 (52–95)	53	Any AML	VEN+HMA	17
Begna, 2021 [30]	1	28	75 (71–92)	nr	Any AML	VEN+HMA	24
Cherry, 2021 [31]	1	143	70 (22–91)	50	Any AML	VEN+AZA	83
De Bellis, 2022 [21]	8	51	75 (55–82)	52	Any AML	VEN+HMA	33
Feld, 2021 [32]	1	26	72	64	Any AML	VEN+HMA	31
Fleishmann, 2022 [33]	1	17	67 (34–83)	57	Any AML	VEN+HMA	15
Garciaz, 2022 [34]	1	39	73 (61–81)	55	Any AML	VEN+AZA	15
Kwang, 2022 [35]	1	74	71	43	Any AML	VEN+DEC	48
Matthews, 2022 [27]	285	129	nr	nr	Any AML	VEN+AZA	33
Mirgh, 2021 [36]	1	24	60 (30–79)	46	Any AML	VEN+HMA	26
Morsia, 2020 [37]	1	44	74 (37–91)	61	Any AML	VEN+HMA	23
Mustafa Ali, 2022 [38]	1	51	nr	59	Any AML	VEN+HMA	28
Salhotra, 2021 [39]	1	30	63 (35–72)	46	Secondary AML	VEN+HMA	29
Shah, 2022 [40]	4	32	72 (61–75)	54	Therapy-related AML	VEN+HMA	26
Todisco, 2023 [24]	32	43	74	47	Any AML	VEN+HMA	24
Vachhani, 2022 [26]	280	169	77 (39–85)	56	Any AML	VEN+HMA	31
Venugopal, 2021 [41]	1	58	73 (26–85)	48	TP53-mutated AML	VEN+HMA	43
Winters, 2019 [42]	1	30	72	nr	Any AML	VEN+AZA	13
DiNardo, 2020 [14]	134	286	76 (49–91)	60	Any AML	VEN+AZA	31

AML: acute myeloid leukemia; VEN+HMA: venetoclax plus hypomethylating agent; VEN+AZA: venetoclax plus azacitidine; VEN+DEC: venetoclax plus decitabine; nr: not reported for the considered subgroup.

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
