# Peer review of "Venetoclax with Hypomethylating Agents in Newly Diagnosed Acute Myeloid Leukemia: A Systematic Review and Meta-Analysis of Survival Data from Real-World Studies"

_cancers, 2023, doi:10.3390/cancers15184618_

Round 1
Reviewer 1 Report
In this paper, "Venetoclax with hypomethylating agents in newly diagnosed acute myeloid leukemia: a systematic review and meta-analysis of real-world studies", Ucciero and colleagues conducted a systematic review and meta-analysis to compare the overall survival of patients with newly diagnosed AML after treatment with venetoclax (VEN) and hypomethylating agents, specifically azacytidine, between a large phase 3 clinical trial, VIALE-A, and the 19 real-world clinical trials. The manuscript is generally well-written and well-organized. The reviewer believes that the results of this study would provide important data for clinicians. However, several issues should be addressed. My specific comments on the manuscript are as follows:
[Major comments]
1) The authors should revise Table 1 to accurately reflect the reported data and improve readability.
a) The authors reported "Males %" of the study by Aiba et al. (Ref. 16) as "nr". However, the number of males was described in the "Results" section of the cited paper as "A total of 13 patients were identified, 10 men and 3 women". According to this update, the percentage of males should be revised (line 148).
b) Please correct the " Age " of the study by Begna et al. (Ref. 18). I found that the median age and range were 75 and 71-92, respectively, in the cited paper.
c) Please correct the "N. Patients" and "Age" of the study by Shah et al. (Ref. 28). I found that the number of patients with treatment-related AML in the cited paper was 32, and the median age was 72.
d) The "Study design" column should be revised to improve readability because all the studies included in this analysis were real-world studies, as stated in "Inclusion and exclusion criteria (line 91)". Even if this column were omitted, the reader would still understand that all of the studies except the one by DiNardo et al. were conducted in real-world settings.
e) Please correct the information on the study by Matthews et al. (Ref. 14). They used data collected from two different databases. One provides data collected from 5 hospitals, and the other provides the de-identified data originated from approximately 280 cancer clinics (~800 sites of care).
f) Please carefully recheck other Tables as well as Table 1 to see if the information was cited correctly.
2) Please change the title to clarify that this is a meta-analysis of the survival data.
3)Please correct lines 52-58 of the Introduction. I needed help understanding the meaning of the sentence.
[Minor comments]
1) Is the statement on lines 60-61 regarding IDH1/2 and hedgehog inhibitors necessary?
2) Please provide a specific period, not "a dramatic increase in survival" (lines 68-69).
3) The VIALE-A study was published in August 2020; why was the start date of the literature search set to January 2019?
4) I could not reproduce the PubMed search result. Although the authors reported 724 records on PubMed, I found 730 records. Please recheck the search results.
5) Please spell out the term "ELN" (line 107).
6) Please correct the typo "mg/die" in lines 224-225.
Some errors should be corrected (please see the minor comments).
Reviewer 2 Report
This is a very interesting systemativ review and meta analysis that deals with a very important topic that is of high relevance for the whole AML comunity. It is very well written and the results are presented well. I have a few suggestions:
1) VIALE-A analyzen VEN/AZA. In this review, the pooled analyses og HMA/VEN are worse than the VIALE-A in terms of OS. However, when they focussed on VEN/AZA, it was not worse than the VIALE-A (Figure 3). Can the authors comment on that? Can you really conclude that VEN/AZA in the real-world is worse than in the VIALE-A? Can they conclude that VEN/AZA is better than VEN/DEC? Discussion of this data would really improve the paper. Could they maybe also include the data about the randomized VEN/LDAC study in this OS comparison (the latter only if it is easily possible..)
2) Please include p-values in the pairwise comparisons in figure S2.
3) Is there any information axceeding OS in the analyzed papers? Is it possible to compare CR rates?
